# Tight Asymptotics of Extreme Order Statistics

**José R. Correa**
Universidad de Chile
correa@uchile.cl

**Frederik Mallmann-Trenn**
King's College London
frederik.mallmann-trenn@kcl.ac.uk

**Matías Romero**
Columbia University
mer2262@columbia.edu

## Abstract

A classic statistical problem is to study the asymptotic behavior of the order statistics of a large number of independent samples taken from a distribution with finite expectation. This behavior has implications for several core problems in machine learning and economics — including robust learning under adversarial noise, best-arm identification in bandit algorithms, revenue estimation in second-price auctions, and the analysis of tail-sensitive statistics used in out-of-distribution detection.

The research question we tackle in this paper is: How large can the expectation of the $\ell$-th maximum of the $n$ samples be? For $\ell = 1$, i.e., the maximum, this expectation is known to grow as $o(n)$, which can be shown to be tight. We show that there is a sharp contrast when considering any fixed $\ell > 1$. Surprisingly, in this case, the largest possible growth rate for all fixed $\ell > 1$ is $O(\frac{n}{\log(n)\log\log(n)})$ and $\Omega(\frac{n}{\log(n)(\log\log(n))^{1.01}})$. Our result is actually finer than the latter and provides a sharp characterization of the largest achievable growth rate for the expectation of the $\ell$-th maximum of $n$ i.i.d. samples.

Beyond the theoretical analysis, we support our findings with extensive simulations. These empirical results highlight a notable phenomenon: although the multiplicative gap between the maximum and the second maximum grows quickly with $n$, the ratio remains approximately constant in 99% of trials. This suggests that while worst-case growth is sharp and meaningful, typical-case behavior may be significantly more stable.

## 1 Introduction

A fundamental problem in applied probability and statistics is to estimate the distribution of the maxima and minima of a set of independent and identically distributed random variables. Specifically, the setting we study in this paper is the following. Let $X_1, X_2, \ldots, X_n$ be non-negative independent and identically distributed (i.i.d.) random variables drawn from a distribution $F$. The corresponding order statistics are:

$$\min_{i \in [n]} X_i = X_{1:n} \leq X_{2:n} \leq \cdots \leq X_{n:n} = \max_{i \in [n]} X_i.$$

Our object of interest is $M_\ell(n) = M_\ell^F(n) = \mathbb{E}\left[X_{n-\ell+1:n}\right].$[1] Our goal is to study the asymptotics of $M_\ell(n)$. In particular, we are interested in the largest possible rate at which $M_\ell(n)$ can grow.

---

[1] When clear from context we will drop the dependence on $F$.

39th Conference on Neural Information Processing Systems (NeurIPS 2025).

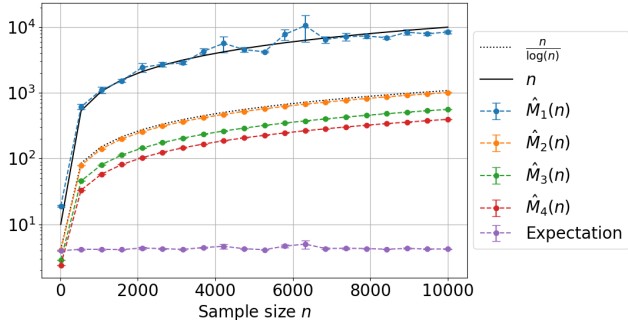

Figure 1: Simulation of expected extremes for a distribution with finite expectation but infinite variance (see Section 5 for more details).

Interestingly, while the expectation of the maximum sample, $M_1(n)$, has been widely studied, the situation is radically different for $M_\ell(n)$ with $\ell > 1$. The first obvious bound follows since the maximum is upper bounded by the sum and thus $M_1(n) \leq n\mathbb{E}[X_1]$, so that, if we have a finite expectation distribution, we get that $M_1(n) = O(n)$. Furthermore, tighter bounds have long been studied assuming more constraints of the distribution such as finite moments (e.g.,[3] and [5]). The bound of $M_1(n) = O(n)$ is actually achievable if the distribution $F$ can depend on $n$, however when $F$ is fixed and only $n$ grows, a stronger bound can be obtained: [10] established that $M_1(n) = o(n)$. Furthermore, Downey proves that this bound is essentially the best possible in the sense that for all $\varepsilon > 0$ there exists a distribution $F$, such that $M_1(n) = \Omega(n^{1-\varepsilon})$. Recently, [8] strengthened the lower bound by showing that for any sublinear function $g(n)$, there exists a finite expectation distribution $F$ such that $M_1(n) = \Omega(g(n))$. This completes the picture for the asymptotic behaviour of $M_1(n)$; it holds that $M_1(n) = o(n)$, but for any function $g(n) \in o(n)$ there exists a finite expectation distribution $F$ such that $M_1^F(n) \geq g(n)$.

In this paper, we explore the asymptotic behavior of the second maximum of $n$ i.i.d. samples and, more generally, $M_\ell(n)$ for fixed $\ell > 1$. The natural reaction here is that the asymptotic behavior of $M_2(n)$ should be essentially the same as that of $M_1(n)$. However, we show a surprisingly stark difference and that the comparison between the first and second maxima is more delicate (see Figure 1). On the one hand, we note that for any sublinear function $g(n)$, there exists a distribution $F$ with finite expectation such that $M_2^F(n) \geq g(n)$ for infinitely many values of $n$. In other words, we show that $\limsup_n M_2^F(n)/g(n) > 0$. On the flip side, if we want to have the inequality for all $n$, then we prove that no such distribution exists. This involves studying the $\liminf_n M_2(n)/g(n)$, which is positive if and only if the $M_2(n) = \Omega(g(n))$.[2] Perhaps surprisingly, our results imply that if we take $g(n) = \frac{n}{\log(n)\log\log(n)}$, then for any distribution $\liminf_n M_2(n)/g(n) = 0$, whereas for $g(n) = \frac{n}{\log(n)(\log\log(n))^{1.01}}$ there exists a finite expectation distribution such that $\liminf_n M_2(n)/g(n) > 0$. More precisely, in Theorem 2 we obtain a tight characterization that generalizes to the $\ell$-th largest sample for any fixed $\ell$. Our necessary condition to have $M_\ell^F(n) = \Omega(g(n))$ generalizes easily for distributions with higher moments: if $F$ has finite variance, in particular, $M_\ell^F(n) = \Omega\left(\sqrt{\frac{n}{\log n}}\right)$ is impossible.

## 1.1  Additional Related Work

In the asymptotic case of order statistics the goal is to study the properties of $X_{k:n}$ as $n$ tends to infinity. A plethora of results are known for asymptotic distributions, theory of extremes and even extremal processes, which have been developed at length in [11]. Their results depend on $p = \lim_{n\to\infty} k/n \in [0, 1]$ and they study three categories ([9]): (1) *Central or quantile case:* If $p \in (0, 1)$, (2) *Extreme case:* If $p \in \{0, 1\}$, for fixed $k$. $X_{k:n}$ and $X_{n-k+1:n}$ are usually called the $k$-th lower and upper extremes, respectively, (3) *Intermediate case:* If $p \in \{0, 1\}$, with $k = k(n) \to \infty$. The scope of our work lies in the second case.

---

[2]Interestingly, $\limsup_n M_2(n)/g(n)$ and $\liminf_n M_2(n)/g(n)$ may well be different.

Results that study the asymptotic behavior of the expected maximum have been further studied in [2], who show that if $\mathbb{E}\left[X^p\right] < \infty$, then the expected maximum $M_1(n) \leq \mathbb{E}\left[X\right] + n^{\frac{1}{p}}\mathbb{E}\left[|X - \mathbb{E}\left[X\right]|^p\right] = O(n^{\frac{1}{p}})$, and for each $n$ the bound is achievable by an extremal distribution $F_n$. [5] give bounds on the expected $k$-th order statistic (i.e., $\mathbb{E}\left[X_{k:n}\right] = M_{n-k+1}(n)$) using information about the first and second moment improving the work of [4] and of [17]. When $F$ is fixed and does not depend on $n$ a much stronger and general bound can be obtained. Indeed, [10] shows that $M_1^F(n) = o(n)$.[3] An alternative and simpler proof of Downey's result was recently given by [8]. They also prove an impossibility result showing that the $o(n)$ bound cannot be improved. Specifically, they showed that for any function $g$ with sublinear growth, namely such that $g(n) = o(n)$, there is a finite expectation distribution $F$ such that $M_1^F(n) = \Omega(g(n))$.

**Applications**

While our primary motivation is theoretical, the asymptotic behavior of extreme order statistics plays a fundamental role in several areas of machine learning and economics. Below, we highlight key settings where our results yield insights.

1. **Robust Learning and Adversarial Risk.** In robust learning, models are often evaluated under worst-case scenarios, such as adversarial examples or corrupted data. Rather than minimizing the average loss, some approaches focus on the largest or top-$\ell$ losses across training examples. Our tight bounds on the $\ell$-th largest sample provide a principled understanding of how such tail-sensitive objectives scale with the number of examples, especially when the underlying distribution has heavy tails [14].

2. **Best-Arm Identification in Bandits.** In stochastic multi-armed bandit problems, particularly in pure exploration settings, identifying the top-performing arms often involves analyzing the largest observed rewards. The expectation of the $\ell$-th maximum is directly tied to evaluating near-optimal arms [13].

3. **Second-Price Auctions.** In second-price (Vickrey) auctions, the seller's revenue is determined by the second-highest bid. When the distribution of bidders' values is unknown or adversarially chosen, our results provide tight upper bounds on the expected revenue a seller can achieve. This supports worst-case analysis in auction design and recent approaches for detecting collusion or setting reserve prices in data-agnostic ways [7, 1].

4. **Out-of-Distribution (OOD) Detection.** Tail-sensitive scoring functions—such as maximum softmax scores or energy-based confidence measures—are widely used for detecting out-of-distribution inputs in classification models [12, 15]. While our work does not propose new detection methods, it offers theoretical insights into how the extreme scores behave as the sample size grows. For example, if the second larges value appears to grow faster than $n/(\log n \log\log n)$, then it's likely an outlier. This may aid in understanding failure modes and calibrating thresholds in high-dimensional or heavy-tailed settings.

5. **Stability of Random Forests.** The sensitivity of the output of a machine learning algorithm to the removal of a single training point—commonly referred to as *algorithmic stability*—is widely studied in the machine learning literature [6, 18]. Recent work by [19] establishes the stability of random forest predictors in regression tasks by showing that it follows from the sublinear growth of the expected maximum of the squared response, a condition satisfied under finite variance. Our bounds on the $\ell$-th largest order statistic may support the analysis of stability guarantees for heavy-tailed responses, where *clipping* the prediction outcome to some data-dependent range, defined by extreme order statistics, can help control the scale of the model's outputs [18].

## 1.2 Overview of our Results

Our main result is to obtain a tight characterization of the $\liminf_n M_\ell^F(n)/g(n)$ for any finite expectation distribution $F$. Recall that for any $F$, $M_1^F(n) = o(n)$, so that we have $\liminf_n M_1^F(n)/n = 0$, and we also know that for any sublinear $g(n)$, there are $F$'s for which $\liminf_n M_1^F(n)/g(n) > 0$ ([10, 8]).

---

[3]More generally Downey establishes that if $\mathbb{E}\left[X_1^p\right] < \infty$, then $M_1^F(n) = o(\sqrt[p]{n})$.

One would expect that the situation for the second, or third maximum would be the same. However, we show that the behavior changes radically. While it is easy to observe that for any sublinear $g(n) = o(n)$, there exists a finite expectation $F$ such that $M_\ell^F(n) \geq g(n)$ infinitely often for any constant $\ell$[4], when looking at the actual growth rate (given by the $\liminf_n M_\ell^F(n)/g(n)$) this *threshold* function $g(n)$ changes dramatically.

In Theorem 2 we show that for $\ell \geq 2$, there exists a finite expectation distribution $F$ such that $M_\ell^F(n) = \Omega(g(n))$ if and only if $\sum_n \frac{g(n)}{n^2} < \infty$. This means that if $g(n)$ satisfies the previous condition,[5] then there are finite expectation distribution $F$ for which $\liminf_n M_\ell^F(n)/g(n) > 0$. However, if, on the contrary, $\sum_n \frac{g(n)}{n^2} = \infty$,[6] then $\liminf_n M_\ell^F(n)/g(n) = 0$ for any finite expectation distribution $F$. As a result, the growth rate of the $\ell$-th maximum cannot be larger than $n/(\log n \log \log n)$ (see Section 4 for precise bounds).

A key feature of our analysis is a simple transformation of any finite expectation distribution $F$ into a distribution $F^+$ which takes the form of an *escalator*. This is done in Section 2. Then, our analysis consists of two keys parts. First, in Theorem 1 (Section 3) we prove asymptotically tight bounds on the growth rate of $M_\ell^F(n)$, for any distribution with finite expectation. Then, in Theorem 2, presented in Section 4, we establish the aforementioned condition by using the bounds of Theorem 1 and an averaging argument over all values of $n$.

**Empirical Validation and Observations.** We complement our theoretical findings with extensive numerical experiments. While our analysis centers on the true expected order statistics $M_\ell(n)$, our empirical simulations shed light on the behavior of their empirical estimates across repeated trials $\hat{M}_\ell(n)$. and support the theoretical results in several ways.

As anticipated, for some distributions, the ratio $\hat{M}_1(n)/\hat{M}_2(n)$ increases quickly as a function of $n$. Even with a large number of trials and high values of $n$, the ratio exhibits significant variability—an expected consequence of the infinite variance of the distribution used. Interestingly, this heavy-tailed behavior appears to predominantly impact $\hat{M}_1(n)$, while $\hat{M}_2(n)$ and lower order statistics remain concentrated.

An interesting phenomenon emerges when we censor the most extreme values: after removing the top 1% of trials with the largest maxima, the ratio $\hat{M}_1(n)/\hat{M}_2(n)$ stabilizes and remains nearly constant across values of $n$. This same near-constant behavior of the ratio is observed when using distributions with finite variance, suggesting that the instability of the ratio is largely driven by rare but extreme deviations in the maximum.

## 2 Reducing to Escalator Distributions

Given $n$ samples $X_1, X_2, \ldots, X_n$ drawn independently (i.i.d.) from a distribution $F$, we let $M_\ell^F(n)$ for $\ell \in \{1, \ldots, n\}$ denote the expected value of the $\ell$-th largest sample.

Given a general distribution $F$, our goal in this section is to show that the asymptotic behavior of $M_\ell^F(n)$ is unchanged if we consider a distribution related to $F$, with convenient structural properties which we call *escalator distribution*. The family of escalator distributions only take values $1 - 1/k$ for integer $k$. An example is the following. For all natural numbers $k$, fix the interval $I_k = [k-1, k)$, in which the distribution (CDF) $F$ takes value $1 - 1/k$. This defines a discrete random variable $X$ on the natural numbers such that $\mathbb{P}[X = k] = F(k+1) - F(k) = 1/(k(k+1))$.

In order to formally define escalator probability distributions, let us consider the indicator function of a subset of real numbers $I$ defined as

$$\mathbf{1}_I(x) = \begin{cases} 1 & \text{if } x \in I \\ 0 & \text{otherwise} \end{cases}.$$

**Definition 1.** *We say a distribution $F^*$ over non-negative reals is an* escalator distribution *if for all $x \geq 0$, $F^*(x) \in \{(1 - 1/k) \mid k \in \mathbb{N}\}$. For any distribution $F$ (not necessarily an escalator*

---

[4]That is, $\limsup_n M_\ell^F(n)/g(n) > 0$.

[5]As it is the case for $g(n) = n/(\log n (\log \log n)^{1.01})$.

[6]As it is the case for $g(n) = n/(\log n \log \log n)$.

*distribution), we define the two associated* escalator distributions $F^+, F^-$ *as follows: For $k \geq 1$ define $x_k = \inf\{x : F(x) \geq 1 - 1/k\}$, and*

$$F^+(x) = \sum_{k=1}^{\infty} \left(1 - \frac{1}{k}\right) \boldsymbol{I}_{[x_k, x_{k+1})}(x), \quad F^-(x) = \sum_{k=1}^{\infty} \left(1 - \frac{1}{k}\right) \boldsymbol{I}_{[x_{k-1}, x_k)}(x).$$

*We define the* step sizes $\delta_k = x_{k+1} - x_k \geq 0$. *In this case, if $X$ is drawn from $F^+$, then $\mathbb{E}[X] = \sum_k \delta_k/k$. If $X$ is drawn from $F^-$, then $\mathbb{E}[X] = \sum_k \delta_{k-1}/k$.*

Note that $F^+(x), F^-(x)$ are CDFs since they are monotonically increasing and converge to 1 as $x$ tends to infinity. It is clear that $F^+ \leq F \leq F^-$ and therefore[7]

$$M_\ell^{F^-}(n) \leq M_\ell^F(n) \leq M_\ell^{F^+}(n). \tag{1}$$

Therefore, instead of studying $F$, we can study the associated escalator distribution $F^+$ in order to upper bound $M_\ell^F(n)$, since $M_\ell^F(n) \leq M_\ell^{F^+}(n)$. We may simply study the asymptotic growth of $F^+$ so long as it has finite expectation. Next we establish that if $F$ has finite expectation, then so has $F^+$.

**Proposition 1** (Proof in Appendix A.1). *Let $F$ be any distribution with finite expectation, then its associated escalator distribution $F^+$ has finite expectation.[8]*

# 3 Asymptotic Behavior of $M_\ell^F(n)$

In this section, we analyse the asymptotic behavior of the expectation of the $\ell$-th maximum of $n$ i.i.d. samples from a distribution $F$ on the non-negative real numbers. Note that this quantity equals:

$$M_\ell^F(n) = \int_0^\infty (1 - \mathbb{P}[X_{n-\ell+1:n} \leq x])dx = \int_0^\infty \left(1 - \sum_{i=0}^{\ell-1} \binom{n}{i} F^{n-i}(x) \cdot (1 - F(x))^i\right) dx.$$

Our main result in this section is the following.

**Theorem 1** (Proof in Appendix A.5). *For any distribution $F$, for any integer $\ell \geq 1$, we have that*

$$M_\ell^F(n) = \Theta\left(\sum_{k=1}^{n-1} \delta_k + \sum_{k=n}^{\infty} \frac{n^\ell}{k^\ell} \delta_k\right),$$

*where $\delta_k$ is defined in Definition 1.*

We prove this theorem in four steps. First, we consider the associated escalator distribution $F^+$ and transform $M_\ell^{F^+}(n)$ into a weighted-sum of the $\delta_k$s (Proposition 2). Second, we derive asymptotically tight (up to constant) bounds (Lemma 3), for $k \leq n$. Third, in Lemma 4, we derive bounds for the case $k > n$. Finally, in the proof of the theorem, we combine all of the above and show that $M_\ell^F(n) = \Theta\left(M_\ell^{F^+}(n)\right) = \Theta\left(\sum_{k=1}^{n-1} \delta_k + \sum_{k=n}^{\infty} \frac{n^\ell}{k^\ell} \delta_k\right)$.

**Proposition 2** (Proof in Appendix A.2). *Let $Y_k \sim \text{BINOMIAL}(n, 1/k)$, we have that*

$$M_\ell^{F^+}(n) = \sum_{k=1}^{\infty} \mathbb{P}[Y_k \geq \ell] \delta_k, \quad M_\ell^{F^-}(n) = \sum_{k=1}^{\infty} \mathbb{P}[Y_{k+1} \geq \ell] \delta_k.$$

Intuitively, the binomials $Y_k$ appear here for the following reason. In order for $X_{n-\ell+1:n}$ to be in the interval $\mathcal{I}_k$, it is necessary that at least $\ell$ variables are in the intervals $\mathcal{I}_k, \mathcal{I}_{k+1} \ldots$, where $\mathcal{I}_i$ are the intervals associated with the $\delta_i$ (see Definition 1). With Proposition 2 equipped, we are ready to prove the following lemma.

---

[7]This is a straight-forward extension of ([16]) who prove that if $F^+ \leq F$ then $M_\ell^F(n) \leq M_\ell^{F^+}(n)$.

[8]As an example, consider the exponential distribution with parameter $\lambda$, $F(x) = 1 - e^{-\lambda x}$. Then, $F^+$ is defined by the intervals $I_k = (\ln(k)/\lambda, \ln(k+1)/\lambda]$, and defines a discrete random variable $Y$ such that $\mathbb{E}[Y] = \frac{1}{\lambda} \sum_k \ln(1 + 1/k)/k < \infty$.

**Lemma 3** (Proof in Appendix A.3)**.** *Let $Y_k \sim \textsc{Binomial}(n, 1/k)$. Let $\ell$ be a fixed constant. For $n \geq 2e\ell! + 1$ and $k \leq n$ we have that,*

$$\mathbb{P}\left[Y_k \geq \ell\right] \in \left[\frac{1}{2e}\frac{1}{\ell!}, 1\right].$$

We now prove the following lemma, which bounds the partial sum of Theorem 1 for $k > n$.

**Lemma 4** (Proof in Appendix A.4)**.** *Let $Y_k \sim \textsc{Binomial}(n, 1/k)$. Let $\ell$ be a fixed constant. Then, for $n \geq \ell^2/(1 - \sqrt{1 - 1/e})$ and $k > n$ we have that,*

$$\mathbb{P}\left[Y_k \geq \ell\right] \in \left[\frac{e-1}{e^2}\frac{1}{\ell!}\left(\frac{n}{k}\right)^\ell, e\left(\frac{n}{k}\right)^\ell\right].$$

## 4 Main Result

In this section we study the largest possible asymptotic behavior of $M_\ell^F(n)$ using the results of the previous section. As we know from [10], $M_1^F(n)$ is sublinear, but can grow as fast as any sublinear function ([8]). In general for any $\ell$, it is clear that $M_\ell^F(n)$ is sublinear since $M_\ell^F(n) \leq M_1^F(n)$. A natural conjecture is that this is still true that the second (or $\ell$-th) maximum grows as fast as any sublinear function.

Our first observation here says that the answer to this question is more subtle. First, in Proposition 5, we show that for any sublinear function, there are distributions that $M_\ell^F(n)$ surpasses it infinitely often. On the other hand, in our main result Theorem 2, we show that $M_\ell^F(n)$ cannot be consistently above any sublinear function. More precisely, we prove that in order to have a finite expectation distribution with $M_\ell^F(n) = \Omega(g(n))$ it is necessary and sufficient to have $\sum_n \frac{g(n)}{n^2} < \infty$.

**Proposition 5** (Proof in Appendix A.6)**.** *For any non-negative, non-decreasing function $g$ with sublinear growth, i.e. $g(n) = o(n)$, there exists an escalator distribution $F$ with finite expectation such that for infinitely many $n$*

$$M_\ell^F(n) \geq g(n).$$

In other words, the last proposition establish that if $g(n) = o(n)$, then there exists a distribution $F$ such that $\limsup_n M_\ell^F(n)/g(n) > 0$. This result implies that, although $M_\ell^F(n)$ has sublinear growth for any finite expectation $F$, it is impossible to find a specific sublinear function $g$ such that $M_\ell^F(n) = o(g(n))$ for all distributions $F$ with finite expectation.

One might hope that our previous $\limsup$ result could be strengthened to $\liminf$ since for $\ell = 1$, Correa and Romero [8] show that $\liminf_n M_1^F(n)/g(n) > 0$, which translates to $M_1^F(n) = \Omega(g(n))$.

However, we show in Theorem 2 that one cannot hope to extend this result to $\ell \geq 2$ for any sublinear function $g$. In particular, Theorem 2 implies that for any finite expectation distribution there are infinitely many values of $n$ for which $M_\ell^F(n) < \frac{n}{\log n \log \log n}$ (for $\ell \geq 2$).

**Theorem 2** (Proof in Appendix A.7)**.** *Fix an integer $\ell \geq 2$. Let $g(n)$ be a monotonically increasing non-negative function. Then, there exists a distribution $F$ with finite expectation such that $M_\ell^F(n) = \Omega\left(g(n)\right)$ if and only if $\sum_n \frac{g(n)}{n^2} < \infty$.*

The above theorem can also be stated as follows. Consider an increasing function $g(n)$ such that $\sum_{n=1}^{\infty} \frac{g(n)}{n^2} < \infty$, then there is a finite expectation distribution $F$, such that $\liminf_n \frac{M_\ell^F(n)}{g(n)} > 0$. Conversely, if $\sum_{n=1}^{\infty} \frac{g(n)}{n^2} = \infty$, then for each distribution $F$, we have that $\liminf_n \frac{M_\ell^F(n)}{g(n)} = 0$.

Therefore, the big picture is the following. We can always find probability distributions to make $M_\ell^F(n)$ match any sublinear function infinitely often. (for infinitely many $n$). However, if one seeks to find a distribution for which $M_\ell^F(n)$ is large for all $n$, then the threshold lies at $n/\ell_n$ where $\ell_n = \prod_{i=1}^{\infty} \log^{(i)}(n)$, where for all $i \in \mathbb{N}$ and $n \in \mathbb{R}^+$

$$\log^{(i)}(n) = \begin{cases} \log(\log^{(i-1)}(n)) & \text{for } i \geq 2 \text{ and } \log^{(i-1)}(n) \geq 2 \\ 1 & \text{for } i \geq 2 \text{ and } \log^{(i-1)}(n) < 2 \\ \log(n) & \text{for } i = 1. \end{cases} \quad (2)$$

In particular, $M_\ell^F(n) = \Omega(\frac{n}{\log n \log \log^{1+\varepsilon} n})$ with $\varepsilon > 0$ is achievable, whereas $M_\ell^F(n) = \Omega(\frac{n}{\log n \log \log n})$ is not.

The construction of finite expectation distributions with large growth rate of $M_\ell^F(n)$ intuitively depends on the tail being heavy enough. If higher moments exist, order statistics grow much slowly. This idea has been established in the literature when deriving upper bounds for the expected maximum ([10],[5],[8]). Similarly, one implication from Theorem 2 can be extended in the same way as follows. Suppose $\mathbb{E}[X^p] < \infty$ for $p > 1$. Since power functions are convex and non-decreasing, Jensen's inequality imply

$$(M_\ell^F(n))^p = (\mathbb{E}[X_{n-\ell+1:n}])^p \leq \mathbb{E}\left[X_{n-\ell+1:n}^p\right] = \mathbb{E}[Z_{n-\ell+1:n}] = M_\ell^{\tilde{F}}(n),$$

where $Z_i = X_i^p$ are non-negative i.i.d. random variables with distribution $\tilde{F}$. Thus, Theorem 2 states that if $g(n)$ satisfies $\sum_n \frac{g(n)}{n^2} = \infty$, then for $\ell \geq 2$

$$\liminf_n \frac{M_\ell^F(n)}{\sqrt[p]{g(n)}} \leq \sqrt[p]{\liminf_n \frac{M_\ell^{\tilde{F}}(n)}{g(n)}} = 0.$$

In particular, $M_\ell^F(n) = \Omega\left(\sqrt{\frac{n}{\log n}}\right)$ is not achievable by any $F$ with finite variance. We establish this generalization in the following corollary.

**Corollary 1.** *Fix $p > 1$ and an integer $\ell \geq 2$. Let $g(n)$ be a monotonically increasing non-negative function such that $\sum_n \frac{g(n)^p}{n^2} = \infty$. Then, any distribution $F$ with $\mathbb{E}[X^p] < \infty$ satisfy $\liminf_n M_\ell^F(n)/g(n) = 0$.*

## 5    Experiments

We complement our theoretical results with simulation experiments that illustrate how the separation between $M_1(n)$ and $M_2(n)$ manifests itself in practice. Our goal is twofold: (i) to empirically validate our results even for moderate values of $n$, and (ii) to explore whether this gap comes from many observations where the maximum sample is consistently larger than the second-highest sample, or from rare events where the separation is extremely large.

### 5.1    Experimental Setup

We focus on *heavy-tailed* distributions with finite expectation, considering two cases: one with **infinite-variance** tail behavior and another with **finite-variance** tail behavior.

We construct each distribution by defining its *tail-quantile function* $T(y) = \inf\{x : F(x) \geq 1 - 1/y\}$. Both distributions were sampled efficiently via inverse transform sampling using a PARETO(1) auxiliary variable. For each distribution, we generate 1,000,000 independent trials for values of $n$ ranging from 10 to 10,000. In each trial, we computed the top order statistics $X_{n:n}, X_{n-1:n}, X_{n-2:n}, X_{n-3:n}$ and averaged across trials to estimate $\hat{M}_\ell$. We focus in particular on the ratio $\hat{M}_1/\hat{M}_2$ as a proxy for the separation between the top two samples.

All experiments were conducted on the Columbia Business School (CBS) Research Grid, a high-performance computing cluster running a Linux environment (Debian 4.19). We fixed the random seed to 42 using Python's default pseudo-random number generator to ensure reproducibility. The code is included in the supplementary material.

### 5.2    Separation between $\hat{M}_1(n)$ and $\hat{M}_2(n)$ under Heavy-Tailed Distributions

We consider $g(n) = n/\log(n)$ as the benchmark growth rate, since Theorem 2 implies that no distribution with finite expectation $F$ can achieve $M_2^F(n) = \Omega(n/\log(n))$. We consider a heavy-tailed distribution defined by the tail-quantile function

$$T(y) = \frac{y}{\log(y)^{1.01}},$$

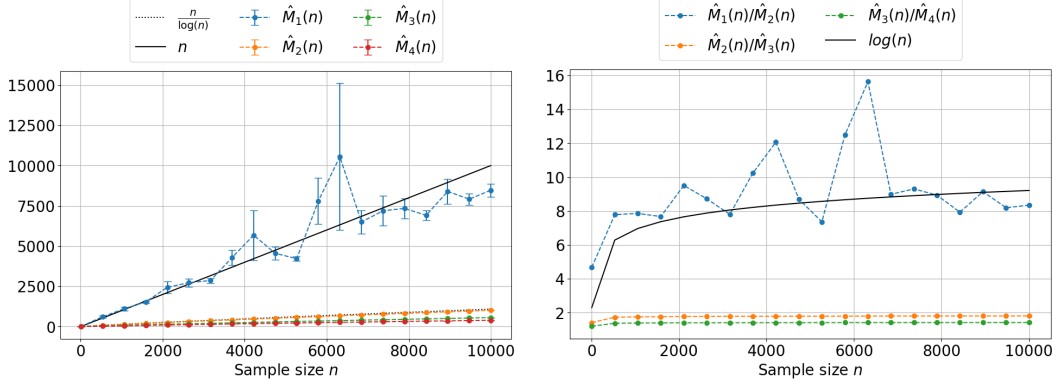

(a) Expected order statistics under infinite-variance distribution. We report the average and standard error over 1,000,000 trials.

(b) Ratios between consecutive expected order statistics (point estimates).

Figure 2: Empirical behavior under $T(y) = y/\log(y)^{1.01}$. $\hat{M}_1(n)$ grows rapidly with large variance; $\hat{M}_2(n)$ and lower order statistics grow more slowly and are tightly concentrated.

which has finite mean but infinite variance. We observe that the ratio $\hat{M}_1(n)/\hat{M}_2(n)$ increases rapidly with $n$. While $\hat{M}_1(n)$ grows nearly linearly, $\hat{M}_2(n)$ remains close to the benchmark growth rate $n/\log n$. Figure 2a shows the estimated expectations of $M_\ell(n)$ for $\ell = 1, 2, 3, 4$. Notably, $\hat{M}_1$ is highly variable even across $10^6$ trials, due to its infinite variance. In contrast, $\hat{M}_2$ and lower order statistics are sharply concentrated. This is explained by the fact that the second moment of $X_{n:n}$ is infinite under this distribution. Indeed, if $X_i = T(V_i)$ with IID samples $V_i \sim \text{PARETO}(1)$, then

$$
\begin{aligned}
\mathbb{E}\left[X_{n:n}^2\right] &= \int_1^\infty n \frac{T(y)^2}{y^2} \left(1 - \frac{1}{y}\right)^{n-1} dy \\
&\geq n \int_2^\infty \frac{T(y)^2}{y^2} \left(1 - \frac{1}{y}\right)^{n-1} dy \\
&\geq \frac{n}{2^{n-1}} \int_2^\infty \frac{T(y)^2}{y^2} dy \\
&= \frac{n}{2^{n-1}} \int_2^\infty \frac{1}{\log(y)^{1+\varepsilon}} dy = \infty.
\end{aligned}
\tag{3}
$$

This divergence explains the persistent noise in estimating $\hat{M}_1(n)$, even with large trial counts.

Figure 2b shows the ratios between consecutive order statistics. We observe a clear upward trend in $\hat{M}_1(n)/\hat{M}_2(n)$, in line with the theoretical expectation that the first and second maxima diverge under infinite variance. Ratios $\hat{M}_2(n)/\hat{M}_3(n)$ and $\hat{M}_3(n)/\hat{M}_4(n)$ remain nearly constant.

### 5.3 Effects of Censoring Extremes

We repeat the comparison after removing the top $1\%$ of trials with the largest maxima. This censoring stabilizes the results dramatically. As shown in Figure 3, the estimates become less noisy, and the ratio $\hat{M}_1(n)/\hat{M}_2(n)$ flattens across $n$. This illustrates that the growth in the ratio is largely driven by rare extreme values of $\hat{M}_1(n)$.

### 5.4 Finite-Variance Comparisons

To test whether this behavior is specific to infinite-variance distributions, we repeat the experiments with a finite-variance distribution defined by

$$
T(y) = \sqrt{\frac{y}{\log(y)^{1.01}}}.
$$

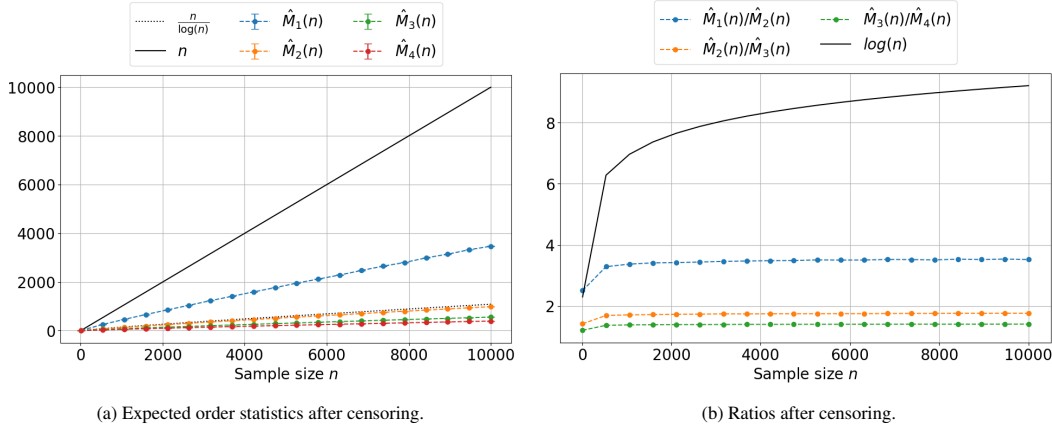

(a) Expected order statistics after censoring.

(b) Ratios after censoring.

Figure 3: Empirical behavior after removing top $1\%$ extreme maxima. The ratio $\hat{M}_1(n)/\hat{M}_2(n)$ becomes stable, suggesting that typical-case gaps are modest.

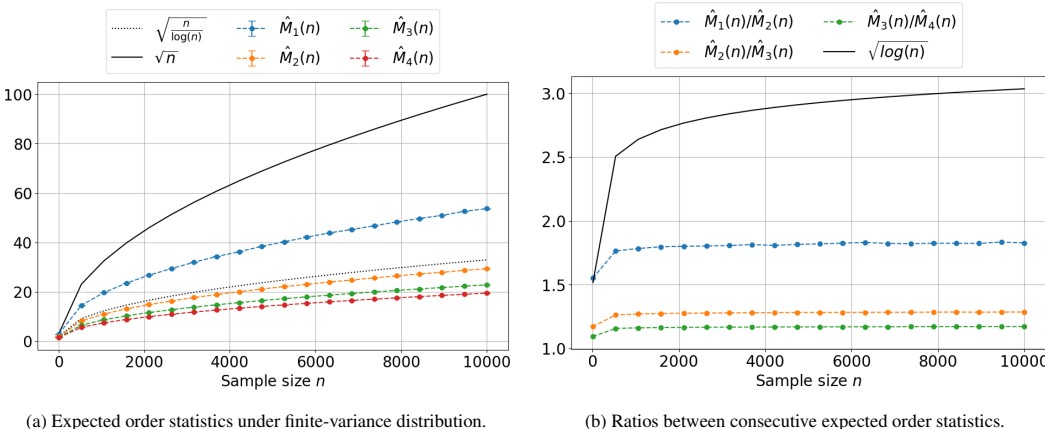

(a) Expected order statistics under finite-variance distribution.

(b) Ratios between consecutive expected order statistics.

Figure 4: Results under $T(y) = \sqrt{y/\log(y)^{1.01}}$. All order statistics are well-behaved and $\hat{M}_1(n)/\hat{M}_2(n)$ remains stable across $n$.

As expected, the results are markedly more stable. We can verify the finite second moment of the maximum:

$$\mathbb{E}\left[X_{n:n}^2\right] = \int_1^\infty n \frac{T(y)^2}{y^2}\left(1 - \frac{1}{y}\right)^{n-1} dy \leq n \int_1^\infty \frac{T(y)^2}{y^2} dy = n \int_1^\infty \frac{1}{y\log(y)^{1.01}} dy < \infty.$$

Hence, the estimates of $\hat{M}_1(n)$ are far less noisy, and the ratio $\hat{M}_1(n)/\hat{M}_2(n)$ remains stable across all tested values of $n$.

## 6 Conclusion and Future Work

In this paper, we have presented asymptotically tight results (up to constants) for $M_\ell(n)$ for any constant $\ell$. The regime where $\ell$ is linear in $n$, comprising the median is well-understood—here, we have that $M_\ell(n) = O(1)$ for any finite expectation distribution. Therefore, a natural open question is what happens in the regime where $\ell \in [\omega(1), o(n)]$. Further enticing open questions are to understand the distribution of $X_{n:n}/X_{n-1:n}$ and to obtain bounds on higher moments of $X_{n-\ell+1:n}$. Finally, it would be interesting to understand the distribution of the second-maximum assuming that higher moments of $F$ are finite. Numerical experiments confirm our theoretical predictions: under heavy tails with infinite variance, the ratio $M_1^F(n)/M_2^F(n)$ can grow with $n$ but does so due to rare extremes. In the typical case—either after censoring or under finite variance—the ratio is stable and bounded.

## Acknowledgments and Disclosure of Funding

The work was in part supported by the EPSRC Grant EP/W005573/1 as well as the Center for Mathematical Modeling (grant ANID Chile FB210005).

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

# A Technical Appendix

## A.1 Proof of Proposition 1

*Proof.* Since $F(x) \leq 1 - \frac{1}{k+1}$ for all $x \in (x_k, x_{k+1}]$, we have that

$$
\begin{aligned}
\int_0^\infty \left( F(x) - F^+(x) \right) dx &= \sum_{k=1}^\infty \int_{x_k}^{x_{k+1}} \left( F(x) - \left( 1 - \frac{1}{k} \right) \right) dx \\
&\leq \sum_{k=1}^\infty \left( \frac{1}{k} - \frac{1}{k+1} \right) (x_{k+1} - x_k) \\
&= \sum_{k=1}^\infty \frac{1}{k(k+1)} (x_{k+1} - x_k) \\
&\leq \sum_{k=1}^\infty \frac{1}{(k+1)} (x_{k+1} - x_k) \\
&= \sum_{k=1}^\infty \int_{x_k}^{x_{k+1}} \left( \frac{1}{(k+1)} \right) dx \\
&\leq \int_0^\infty (1 - F(x)) \, dx \\
&= \mathbb{E}[X_1].
\end{aligned}
$$

Thus,

$$
\mathbb{E}\left[ X_1^+ \right] = \int_0^\infty \left( 1 - F^+(x) \right) dx = \int_0^\infty (1 - F(x)) \, dx + \int_0^\infty \left( F(x) - F^+(x) \right) dx \leq 2\mathbb{E}[X_1] < \infty,
$$

and $F^+$ has finite expectation.

$\square$

## A.2 Proof of Proposition 2

*Proof.* Recall that if $F^+$ is an escalator distribution and $X_1^+, \ldots, X_n^+$ are drawn from $F^+$ with non-negative support, then

$$
M_1^{F^+}(n) = \int_0^\infty \left( 1 - F^{+n}(x) \right) dx = \sum_{k=1}^\infty \left( 1 - \left( 1 - \frac{1}{k} \right)^n \right) \delta_k.
$$

Similarly, using the fact that the distribution of $X_{n-1:n}$ is $F^{+n} + nF^{+n-1}(1 - F^+)$, we get that

$$
\begin{aligned}
M_2^{F^+}(n) &= \int_0^\infty \left( 1 - F^{+n}(x) - nF^{+n-1}(x)(1 - F^+(x)) \right) dx \\
&= \sum_{k=1}^\infty \left( 1 - \left( 1 - \frac{1}{k} \right)^n - n \left( 1 - \frac{1}{k} \right)^{n-1} \frac{1}{k} \right) \delta_k.
\end{aligned}
$$

In general, we get, that $X_{n-\ell+1:n}$ is distributed as $\sum_{i=0}^{\ell-1} \binom{n}{i} F^{+n-i}(x) \cdot (1 - F^+(x))^i$. Thus,

$$M_\ell^{F^+}(n) = \int_0^\infty \left(1 - \sum_{i=0}^{\ell-1} \binom{n}{i} F^{+n-i}(x) \cdot (1 - F^+(x))^i\right) dx$$

$$= \sum_{k=1}^\infty \left(1 - \sum_{i=0}^{\ell-1} \binom{n}{i}\left(1 - \frac{1}{k}\right)^{n-i} \frac{1}{k^i}\right) \delta_k$$

$$= \sum_{k=1}^\infty \left(1 - \sum_{i=0}^{\ell-1} \mathbb{P}\left[Y_k = i\right]\right) \delta_k$$

$$= \sum_{k=1}^\infty \mathbb{P}\left[Y_k \geq \ell\right] \delta_k,$$

concluding the proof for $F^+$. The result for $F^-$ is anologous since its definition only involves shifting $\delta_k$'s indices by one. □

### A.3 Proof of Lemma 3

*Proof.* The upper bound holds trivially, since $\mathbb{P}\left[Y_k \geq \ell\right] \leq 1$. For the lower bound, we have, by Lemma 6, for $k \geq 2$

$$\mathbb{P}\left[Y_k < \ell\right] = \sum_{i=0}^{\ell-1} \binom{n}{i}(1 - 1/k)^{n-i}\frac{1}{k^i}$$

$$= (1 - 1/k)^n \sum_{i=0}^{\ell-1} \binom{n}{i}\frac{1}{(k-1)^i}$$

$$\leq \left(1 - \frac{1}{n}\right)^n \sum_{i=0}^{\ell-1} \binom{n}{i}\frac{1}{(n-1)^i}.$$

Note that for $k = 1$, we have $\mathbb{P}\left[Y_1 < \ell\right] = 0$. Thus, for all $k \geq 1$, we have

$$\mathbb{P}\left[Y_k < \ell\right] \leq \frac{1}{e} \sum_{i=0}^{\ell-1} \binom{n}{i}\frac{1}{(n-1)^i}.$$

Then, since $\ell$ is much smaller than $n$,

$$\sum_{i=0}^{\ell-1} \binom{n}{i}\frac{1}{(n-1)^i} = \sum_{i=0}^{\ell-1} \frac{1}{i!} \prod_{j=0}^{i-1} \frac{n-j}{n-1}$$

$$\leq \left(1 + \frac{1}{n-1}\right) \sum_{i=0}^{\ell-1} \frac{1}{i!}$$

$$\leq \left(1 + \frac{1}{n-1}\right)\left(\sum_{i=0}^\infty \frac{1}{i!} - \frac{1}{\ell!}\right)$$

$$\leq \left(1 + \frac{1}{n-1}\right)\left(e - \frac{1}{\ell!}\right)$$

$$\leq e - \frac{1}{\ell!} + \frac{e}{n-1}.$$

Thus, $\mathbb{P}\left[Y_k \geq \ell\right] = 1 - \mathbb{P}\left[Y_k < \ell\right] \geq 1 - \frac{1}{e}\left(e - \frac{1}{\ell!} + \frac{e}{n-1}\right) = \frac{1}{e\ell!} - \frac{1}{n-1} \geq \frac{1}{2e\ell!}$, for $n \geq 2e\ell! + 1$. □

## A.4 Proof of Lemma 4

*Proof.* Let $Z_k$ be the Poisson distribution with mean $n/k$. Using that $k \geq n+1$, we get,

$$
\begin{aligned}
\mathbb{P}\left[Y_k = i\right] &= \binom{n}{i} \frac{1}{k^i} (1 - 1/k)^{n-i} \\
&\leq \frac{n^i}{i!} \frac{1}{k^i} (1 - 1/k)^{k(n-i)/k} \\
&\leq \frac{(n/k)^i}{i!} e^{-(n-i)/k} \\
&= \mathbb{P}\left[Z_k = i\right] e^{i/k} \\
&\leq \mathbb{P}\left[Z_k = i\right] e^{n/k}.
\end{aligned}
$$

Thus, using that $\sum_{i=\ell}^{\infty} \frac{1}{i!} \leq e$, we get,

$$
\mathbb{P}\left[Z_k \geq \ell\right] = e^{-n/k} \sum_{i=\ell}^{\infty} \frac{(n/k)^i}{i!} \leq e^{-n/k} \cdot (n/k)^\ell \sum_{i=\ell}^{\infty} \frac{1}{i!} \leq e^{1-n/k} \cdot \left(\frac{n}{k}\right)^\ell.
$$

For the lower bound, we use $(1 - 1/k)^k \geq (1 - 1/k)/e$. We first bound,

$$
\begin{aligned}
(1 - 1/k)^{n-\ell} &\geq (1 - 1/k)^{k(n-\ell)/k} & (4) \\
&\geq \left(\frac{1}{e}(1 - 1/k)\right)^{(n-\ell)/k} \\
&\geq \frac{e^{\ell/k}}{e^{n/k}} \left(1 - \frac{n-\ell}{k^2}\right) \\
&\geq \frac{e^{\ell/k}}{e^{n/k}} \left(1 - \frac{1}{n}\right).
\end{aligned}
$$

Thus, since $n \geq \ell^2/(1 - \sqrt{1 - 1/e})$, and using that $\mathbb{P}\left[Z_k = \ell\right] = \frac{(n/k)^\ell e^{-n/k}}{\ell!}$, we get,

$$
\begin{aligned}
\mathbb{P}\left[Y_k = \ell\right] &= \binom{n}{\ell} \frac{1}{k^\ell} (1 - 1/k)^{n-\ell} \\
&\geq \frac{(n-\ell+1)^\ell}{\ell!} \frac{1}{k^\ell} (1 - 1/k)^{n-\ell} \\
&= \mathbb{P}\left[Z_k = \ell\right] \frac{(n-\ell+1)^\ell}{n^i} (1 - 1/k)^{n-\ell} e^{n/k} \\
&\underset{(4)}{\geq} \mathbb{P}\left[Z_k = \ell\right] \left(1 - \frac{\ell-1}{n}\right)^\ell \left(1 - \frac{1}{n}\right) e^{\ell/k} \\
&\geq \mathbb{P}\left[Z_k = \ell\right] \left(1 - \frac{\ell^2}{n}\right) \left(1 - \frac{1}{n}\right) e^{\ell/k} \\
&\geq \mathbb{P}\left[Z_k = \ell\right] \left(1 - \frac{\ell^2}{n}\right)^2 \\
&\geq \mathbb{P}\left[Z_k = \ell\right] (1 - 1/e).
\end{aligned}
$$

Hence, since $e^{-n/k} \geq 1/e$ we get,

$$
\mathbb{P}\left[Y_k \geq \ell\right] \geq \mathbb{P}\left[Y_k = \ell\right] \geq (1 - 1/e)\mathbb{P}\left[Z_k = \ell\right] \geq (1 - 1/e)\frac{1}{e}\frac{1}{\ell!}\left(\frac{n}{k}\right)^\ell.
$$

Establishing the claimed bounds. $\qquad\square$

## A.5 Proof of Theorem 1

*Proof.* From Lemma 3 we get,

$$\frac{1}{2e}\frac{1}{\ell!}\sum_{k=1}^{n}\delta_k \leq \sum_{k=1}^{n}\mathbb{P}\left[Y_k \geq \ell\right]\delta_k \leq \sum_{k=1}^{n}\delta_k.$$

From Lemma 4 we get,

$$\frac{e-1}{e^2}\frac{1}{\ell!}\sum_{k=n+1}^{\infty}\left(\frac{n}{k}\right)^{\ell}\delta_k \leq \sum_{k=n+1}^{\infty}\mathbb{P}\left[Y_k \geq \ell\right]\delta_k \leq e\sum_{k=n+1}^{\infty}\left(\frac{n}{k}\right)^{\ell}\delta_k.$$

Thus, by Proposition 2, for the escalator $F^+$ using that $\sum_{k=1}^{n}\delta_k + \sum_{k=n+1}^{\infty}\frac{n^\ell}{k^\ell}\delta_k = \sum_{k=1}^{n-1}\delta_k + \sum_{k=n}^{\infty}\frac{n^\ell}{k^\ell}\delta_k$, we have

$$M_\ell^{F^+}(n) = \Theta\left(\sum_{k=1}^{n-1}\delta_k + \sum_{k=n}^{\infty}\frac{n^\ell}{k^\ell}\delta_k\right). \tag{5}$$

Similarly, we obtain

$$M_\ell^{F^-}(n) = \Theta\left(\sum_{k=1}^{n-1}\delta_k + \sum_{k=n}^{\infty}\frac{n^\ell}{(k+1)^\ell}\delta_k\right), \tag{6}$$

We claim that if $k \geq n$ and $\ell = O(1)$, then

$$\frac{1}{(k+1)^\ell} \in \left[\frac{1}{2k^\ell}, \frac{1}{k^\ell}\right]. \tag{7}$$

From Eq. (7) together with Eq. (5) and Eq. (6), the claim in the statement of the theorem follows immediately, since for some $c$ large enough,

$$\frac{1}{c}\left(\sum_{k=1}^{n-1}\delta_k + \sum_{k=n}^{\infty}\frac{n^\ell}{k^\ell}\delta_k\right) \leq M_\ell^{F^-}(n) \leq M_\ell^{F}(n) \leq M_\ell^{F^+}(n) \leq c\left(\sum_{k=1}^{n-1}\delta_k + \sum_{k=n}^{\infty}\frac{n^\ell}{k^\ell}\delta_k\right),$$

where we used that $F^+ \leq F \leq F^-$, yielding $M_\ell^{F^-}(n) \leq M_\ell^{F}(n) \leq M_\ell^{F^+}(n)$.

It remains to prove Eq. (7). Clearly $\frac{1}{(k+1)^\ell} \leq \frac{1}{k^\ell}$. For the lower bound, we assume without loss of generality that $\ell \geq 2$ since, otherwise, $\frac{1}{k+1} \geq \frac{1}{2k}$.

We have $\frac{1}{(k+1)^\ell} \geq \frac{1}{2k^\ell}$ since $\frac{k^\ell}{(k+1)^\ell} = (1 - 1/(k+1))^\ell \geq 1/2$. Thus, Eq. (7) holds completing the proof. $\qquad\square$

## A.6 Proof of Proposition 5

*Proof.* For any $i \in \mathbb{N}$, define $n(i) = \min\{n \geq 1 \mid g(n)/n \leq 2^{-i}\}$. Note that $n(i)$ is well-defined, i.e. it is a positive integer, since $g(n) = o(n)$. Moreover, since $g(\cdot)$ is non-decreasing, $n(i) \to \infty$ as $i \to \infty$. Let $\mathcal{I} = \{n(i) \mid i \in \mathbb{N}\}$ and $c = \frac{1}{2e}\frac{1}{\ell!}$. Consider an escalator distribution defined by

$$\delta_k = \begin{cases} \frac{g(k)}{c} & \text{if } k \text{ in } \mathcal{I} \\ 0 & \text{otherwise} \end{cases}$$

Therefore, such escalator distribution has finite expectation. Indeed,

$$\sum_{k=1}^{\infty}\frac{\delta_k}{k} = \sum_{k \in \mathcal{I}}\frac{g(k)}{ck} \leq \frac{1}{c}\sum_{i=1}^{\infty}\frac{1}{2^i} = \frac{1}{c} < \infty.$$

On the other hand, from Proposition 2, we have $M_\ell^{F}(n) \geq \mathbb{P}[Y_n \geq \ell]\delta_n$. Moreover, by Lemma 3, for large enough $n$, $\mathbb{P}[Y_n \geq \ell] \geq c$. Thus, since $n(i) \to \infty$ as $i \to \infty$, for all large enough $i$,

$$M_\ell^{F}(n(i)) \geq c\delta_{n(i)} = g(n(i)).$$

$\qquad\square$

### A.7 Proof of Theorem 2

*Proof.* We start by proving that if $M_\ell^F(n) = \Omega(g(n))$ for some distribution $F$ with finite expectation, then necessarily $\sum_n \frac{g(n)}{n^2} < \infty$. Since $M_\ell^F(n) \leq M_2^F(n)$, it is enough to show the claim for $\ell = 2$. Therefore, assume that for all $n$ we have,

$$M_2^F(n) \geq g(n). \tag{8}$$

By Theorem 1, there exists a constant $c > 0$ such that

$$M_2^F(n) \leq c \sum_{k=1}^{n-1} \delta_k + c \sum_{k=n}^{\infty} \frac{n^2}{k^2} \delta_k =: f(n).$$

From Eq. (8) it follows that $f(n) \geq M_2^F(n) \geq g(n)$. Thus,

$$c \sum_{k=1}^{\infty} \min\left(\frac{1}{n}, \frac{n}{k^2}\right) \delta_k = \frac{f(n)}{n} \geq \frac{g(n)}{n}.$$

Therefore, dividing by $n$ again and summing over all values of $n$ it must hold that

$$c \sum_{n=1}^{\infty} \frac{1}{n} \sum_{k=1}^{\infty} \min\left(\frac{1}{n}, \frac{n}{k^2}\right) \delta_k \geq \sum_{n=1}^{\infty} \frac{g(n)}{n^2}.$$

Switching indices yields

$$\sum_{n=1}^{\infty} \frac{g(n)}{n^2} \leq c \sum_{n=1}^{\infty} \sum_{k=1}^{\infty} \min\left(\frac{1}{n^2}, \frac{1}{k^2}\right) \delta_k$$

$$= c \sum_{n=1}^{\infty} \sum_{k=1}^{\infty} \min\left(\frac{1}{n^2}, \frac{1}{k^2}\right) \delta_n$$

$$= c \sum_{n=1}^{\infty} \frac{\delta_n}{n} \sum_{k=1}^{\infty} \min\left(\frac{1}{n}, \frac{n}{k^2}\right)$$

$$\leq c \sum_{n=1}^{\infty} \frac{\delta_n}{n} \left(\sum_{k=1}^{n} \frac{1}{n} + \sum_{k=n+1}^{\infty} \frac{n}{k^2}\right)$$

$$\leq 3c \sum_{n=1}^{\infty} \frac{\delta_n}{n}.$$

Here we used that $n \sum_{k=n+1}^{\infty} \frac{1}{k^2} = n\Psi(1, n) \leq 2$, where $\Psi$ is the polygamma function. Recall from Proposition 1 that if $F$ has finite expectation, then $\sum_n \frac{\delta_n}{n} < \infty$. Therefore, $\sum_n \frac{g(n)}{n^2} < \infty$.

We now prove the other direction: assuming $\sum_n \frac{g(n)}{n^2} < \infty$, we construct an escalator distribution $F$ with finite expectation such that $M_\ell^F(n) \geq g(n)$. Recall that for any distribution $F$, by Theorem 1, there exists a universal constant $c$ such that $M_\ell^F(n) \geq c \sum_{k=1}^{n-1} \delta_k + c \sum_{k=n}^{\infty} \frac{n^\ell}{k^\ell} \delta_k$. We define $h(n) = n/g(n)$ and construct $F$ by defining

$$\delta_k = \frac{k^\ell}{c} \left(\frac{1}{k^{\ell-1} h(k)} - \frac{1}{(k+1)^{\ell-1} h(k+1)}\right).$$

Indeed, if $X$ is drawn by $F$, using that $\left(1 - \frac{1}{k+1}\right)^{\ell-1} \geq 1 - \frac{\ell-1}{k+1}$, we get,

$$\mathbb{E}[X] = \sum_{k=1}^{\infty} \frac{\delta_k}{k}$$

$$= \frac{1}{c} \sum_{k=1}^{\infty} \left(\frac{1}{h(k)} - \left(1 - \frac{1}{k+1}\right)^{\ell-1} \frac{1}{h(k+1)}\right)$$

$$\leq \frac{1}{c} \sum_{k=1}^{\infty} \left(\frac{1}{h(k)} - \frac{1}{h(k+1)}\right) + \sum_{k=1}^{\infty} \frac{\ell-1}{(k+1) h(k+1)}.$$

The first term evaluates to $1/h(1)$ since it is a telescopic sum and $1/h(k) = g(k)/k \to 0$ as $k \to \infty$, by the assumption that $\sum_{n=1}^{\infty} g(n)/n^2 < \infty$. Similarly, the second sum is finite by hypothesis. Therefore $\mathbb{E}[X] < \infty$, so that the constructed distribution $F$ has finite expectation.

On the other hand, by Theorem 1, we get

$$
\begin{aligned}
M_\ell^F(n) &\geq c \sum_{k=1}^{n-1} \delta_k + c \sum_{k=n}^{\infty} \frac{n^\ell}{k^\ell} \delta_k \\
&\geq n^\ell \sum_{k=n}^{\infty} \left( \frac{1}{k^{\ell-1} h(k)} - \frac{1}{(k+1)^{\ell-1} h(k+1)} \right) \\
&= n^\ell \frac{1}{n^{\ell-1} h(n)} = \frac{n}{h(n)} = g(n).
\end{aligned}
$$

As before, the telescopic sum results only in the first term, because $1/(k^{\ell-1} h(k)) = g(k)/k^\ell \leq g(k)/k \to 0$ as $k \to \infty$. The above implies the existence of a finite expectation distribution $F$ such that $M_\ell^F(n) = \Omega(g(n))$. $\qquad\square$

## A.8 Auxiliary Lemma

The following technical lemma is used to prove Lemma 3.

**Lemma 6.** *For any positive integer $\ell \leq n + 1$, the function*

$$
f(x) = (1 - 1/x)^n \sum_{i=0}^{\ell-1} \binom{n}{i} \frac{1}{(x-1)^i}
$$

*for $x \leq n$ is non-decreasing.*

*Proof.* The derivative w.r.t. $x$ of this function $f(x)$ is

$$
f'(x) = n \left( 1 - \frac{1}{x} \right)^{n-1} \frac{1}{x^2} \sum_{i=0}^{\ell-1} \binom{n}{i} \frac{1}{(x-1)^i} - \left( 1 - \frac{1}{x} \right)^n \sum_{i=1}^{\ell-1} \binom{n}{i} \frac{i}{(x-1)^{i+1}}, \qquad (9)
$$

Equation (9) can be rewritten as

$$
f'(x) = \left( 1 - \frac{1}{x} \right)^{n-1} \frac{1}{x} \left[ \frac{n}{x} + \sum_{i=1}^{\ell-1} \binom{n}{i} \frac{1}{(x-1)^i} \left( \frac{n}{x} - i \right) \right].
$$

Note that all terms in the sum are non-negative for if $x < n/(\ell - 1)$. In the remainder we rewrite the sum and consider the case that $x \geq n/(\ell - 1)$. Since $\binom{n}{i} \frac{i}{n} = \binom{n-1}{i-1}$, we can rewrite the second sum in equation (9) as

$$
\frac{n}{(x-1)^2} \sum_{i=1}^{\ell-1} \binom{n-1}{i-1} \frac{1}{(x-1)^{i-1}} = \frac{n}{(x-1)^2} \sum_{i=0}^{\ell} \binom{n-1}{i} \frac{1}{(x-1)^i},
$$

and then the derivative can be factorized as follows

$$
\left( 1 - \frac{1}{x} \right)^{n-1} \frac{n}{x} \left[ \binom{n}{\ell-1} \frac{1}{x(x-1)^{\ell-1}} + \sum_{i=0}^{\ell-2} \frac{1}{(x-1)^i} \left( \binom{n}{i} \frac{1}{x} - \binom{n-1}{i} \frac{1}{x-1} \right) \right],
$$

where all the terms are non-negative if:

$$
\frac{x-1}{x} \geq \frac{\binom{n-1}{i}}{\binom{n}{i}} = \frac{\binom{n-1}{i}}{\binom{n}{i}} = \frac{\frac{(n-1)!}{i!(n-1-i)!}}{\frac{n!}{i!(n-i)!}} = \frac{n-i}{n}.
$$

Which holds if and only if $\frac{i}{n} \geq \frac{1}{x}$ which in turn is equivalent to $x \geq \frac{n}{i}$. The latter is true since we assume $x \geq n/(\ell - 1)$. Thus, the derivative is non-negative and the claim follows.

$\qquad\square$

