# OpenReview forum: "Tight Asymptotics of Extreme Order Statistics"
_NeurIPS.cc/2025/Conference — NeurIPS 2025 poster_

### Official Review · Reviewer_6S2N · 2025-06-14

**Clarity:** 2
**Significance:** 2
**Originality:** 3
**Rating:** 4
**Confidence:** 2

**Summary:**

This paper studies the behavior of the expected $\ell$-th largest order statistics from $n$ i.i.d. samples (i.e., $(n-\ell+1)$-th order statistic),  focusing on the case $\ell \ge 2$.
The authors show that, for all fixed $\ell \ge 2$, the growth rate of this expected value should be $O(\frac{n}{\log(n)\log\log(n)})$.
Furthermore, they establish a characterization: for a distribution with finite mean, the condition $M_\ell(n) = \Omega(g(n))$ holds if and only if the monotonically increasing sublinear function $g(n)$ satisfies $g(n)$ satisfies$\sum_n g(n)/n^2 < \infty$, where $M_\ell(n)$ denotes the expected $\ell$-th largest order statistics.

**Questions:**

1. If there are any misunderstandings in my assessment of the Strengths and Weaknesses, I would greatly appreciate it if you could point them out. It would be helpful to receive further explanation or clarification where appropriate.
2. Could you elaborate on the distinctions among the three regimes of interest:
(i) the central/quantile case (e.g., the median or fixed-rank quantiles),
(ii) the extreme case (as in this paper, dealing with the largest or near-largest order statistics), and
(iii) the intermediate case?
For example, the different motivations for studying each regime and how results from each are applied or interpreted in practice.

3. I would also appreciate further insight into the contribution of this paper, especially in terms of why the results are theoretically or practically significant. For example, how such tight asymptotics for extreme order statistics (rather than just sample mean) particularly beneficial in best-arm identification examples?

---
### Typos
1. Subsection title 1.2. : overivew -> overview
2. Citation [14]: Daniel Levy and et al. -> need to be revised.
3. In the proof of Proposition 6, missing spaces in Lines 708 and 709: escalatordistribution -> escalator distribution.

**Ethical Concerns:**

["NO or VERY MINOR ethics concerns only"]

**Final Justification:**

My main concerns were the proof of Lemma 4 and the significance of the contribution.

The first point was clarified after reading the authors’ response.

As for the significance, my evaluation is primarily informed by the assessments of the other reviewers, many of whom seem satisfied with the results. However, since I am not familiar with this topic, I remain uncertain about my own evaluation.

**Limitations:**

yes

**Quality:**

2

**Strengths And Weaknesses:**

### Strength

* New theoretical insight for all fixed $\ell\geq 2$.

The paper illustrates how the expected $\ell$-th largest order statistic, $M_\ell(n)$ behaves fundamentally differently from the maximum order statistic. For distribution with the finite expectation, it was shown that $M_1=\Omega(g(n))$ holds for any sublinear $g(n)$ [8], but the authors show that $M_{\ell \geq 2} = \Omega(g(n))$ can hold iff $\sum_n g(n)/n^2 <\infty$.

* Validation of theoretical results by numerical experiments: Simulations cover both finite-variance and infinite-variance cases and track the empirical rate against the theoretical upper– and lower–bound results, giving concrete support to the asymptotic claims.

### Weakness

1. The paper establishes the theoretical behavior of $M_\ell$ that is distinct to $M_1$, but it largely leaves readers on their own to understand what these results mean for concrete examples or for practical tasks. (although there are general applications in Section 1)
Therefore, it would be beneficial to have some discussions on the obtained results.

For example, while the authors explained that it is impossible to derive a specific sublinear function $g(n)$ satisfying $M_\ell^F(n)=o(g(n))$ for given distribution $F$, one may wonder what $M_\ell^F(n)$ looks like for familiar distributions that could be written in the closed formulation beyond $\delta_k$ expression.
In this context it would be useful to illustrate the results with Fréchet-type heavy-tailed distributions, those whose tail quantile function is regularly varying with index $a>0$, i.e. $x_k = a k^a o(k^{\epsilon})$ for any $\epsilon>0$  (see Charras-Garrido & Lezaud, 2013 for more details).
A classical example is the Pareto distribution with shape parameter $b>0$, for which $a=1/b$.
For such tails one can compute explicitly the sublinear order of $M_\ell^F(n)$ making the sub-linear growth far more transparent than the abstract $\delta_k$ expression.
Including such examples would let readers grasp the sub-linear growth rate of $M_\ell^F$ at a glance.

2. The theorem is phrased “for every integer $\ell\geq 1$", but the proof of Lemma 4 (Line 688) assumes $i$ is small enough relative to $n$ to ensure
$$
\left(1-\frac{i^2}{n} \right)\left(1-\frac{1}{n} \right) e^{i/k} \geq (1-1/e), \quad k \geq n+1.
 $$
For example, $i=7, n=122, k=123$ violates the inequality, so we need $n\geq 123$ for fixed $i=7$.
For this reason, I think the statement of Theorem should be restated with an explicit range such as $n \geq n_0(\ell)$ for fixed $\ell$ or for integer $\ell$ small enough relative to $n$.

---
Myriam Charras-Garrido and Pascal Lezaud. Extreme value analysis: an introduction. Journal de la
Soci´et´e Franc¸aise de Statistique, 154(2):66–97, 2013.

---

> ### Author Rebuttal · Authors · 2025-07-31
>
> We thank Reviewer 6S2N for the detailed feedback and questions. Below, we address the concerns raised, clarify points of misunderstanding, and provide additional examples and context where requested.
>
> **1. Concrete examples and practical tasks.**
>
> We appreciate the suggestion to illustrate our results using familiar heavy‑tailed distributions. Indeed, it is instructive to examine the Pareto distribution as a standard example. For Pareto$(1,b)$ with shape parameter $b=1+\epsilon$, the distribution has finite expectation, and it can be shown that $M_1 = \Theta(n^{1/b})$ (see e.g. Downey, ORL 1990). A similar argument shows that $M_2=\Theta(n^{1/b})$. Our work complements such examples by characterizing necessary and sufficient conditions under which the asymptotic behavior of $M_1$ and $M_2$ diverge. Addressing this question required constructing specific distribution families that go beyond well‑known examples, but we agree that further exploring these connections is a natural next step in extending the theoretical boundary of our results.
>
> **2. Assumptions in Lemma 4 and Theorem 1.**
>
> We apologize for the confusion. In Lemma 4, we substitute $ i=\ell $ and do assume that $\ell$ is constant (independent of $n,k$) and that $n$ is large enough with respect to $\ell$, e.g. $n = \Omega(\ell^2)$. We will make this explicit in the statement of Lemma 4 and Theorem 1, consistent with your suggestion.
>
> **3. Distinctions among the three regimes.**
>
> We appreciate the opportunity to clarify the distinction between the three regimes in the asymptotic analysis of order statistics. The central/quantile case is widely used in practice, for example, in risk management to measure the losses in the worst $p$% of scenarios (Value-at-Risk). Theoretically, the asymptotics for $X_{ \lfloor pn \rfloor }$ are well understood since CLT-like results apply under mild assumptions. The extreme case appears naturally in several applications (as mentioned in the paper). Technically, this case differs from the previous one in that the asymptotic distribution is not Gaussian, and additional assumptions need to be made to understand the overall asymptotic behavior (Extreme Value Theory). The fundamental question we address differs from the classic extreme value theory, in that we study the transient behavior as $n$ increases even if a limit distribution may not exist. Finally, the intermediate case is technically more complex than the extreme regime, and we fail to see additional practical motivations for research.
>
> **4. Significance of the contribution.**
>
> We appreciate the opportunity to clarify why the results are impactful. Our paper addresses a simple yet fundamental theoretical question regarding the possible growth rates achievable by extreme order statistics under any distribution with finite expectation. These objects appear naturally in many applications, such as comparing first- or second-price auctions, and our results can be useful in worst-case analysis when the underlying distribution of interest is unknown. The same high-level idea can be potentially used in robust learning, where objectives such as average top‑$\ell$ losses are considered to adapt to imbalanced and/or multi-modal data distributions better than the average loss (Fan et al. NeurIPS 2017). Although further applications remain to be fully explored, our results could be applied more broadly in learning settings where the data‑generating distribution may be adversarially chosen.
>
> References:
>
> Peter J Downey. Distribution-free bounds on the expectation of the maximum with scheduling applications.
> Operations Research Letters, 9(3):189–201, 1990.
>
> Y. Fan, S. Lyu, Y. Ying, and B. Hu. Learning with average top-k loss. In Advances in Neural Information Processing Systems 30, pages 497–505, 2017.

---

> > ### Comment · Reviewer_6S2N · 2025-08-03
> >
> > Thank you for your responses and the effort in addressing the concerns I raised.
> >
> > Most of my concerns have been resolved, and the discussions with other reviewers were also informative. Since I think the current proof is correct, I will update my score accordingly.

---

### Official Review · Reviewer_WsQo · 2025-06-18

**Clarity:** 3
**Significance:** 4
**Originality:** 4
**Rating:** 5
**Confidence:** 3

**Summary:**

This paper studies the behavior of order statistics, and specifically, the expectation of order statistics beyond the maximum. Previous work shows that for all constant distributions the expected maximum of n samples is $o(n)$, and this is tight. This paper studies the growth rate for the \ell-th order statistic. The picture is rather subtle: there exist infinitely many distributions and infinitely many n such that the \ell-th order statistic is an arbitrary sublinear function. However, there are limits to what can be done for a fixed distribution and all n. Specifically, $\Omega(n/(log(n) loglog(n) )$ is not possible for all n, but $\Omega(n/(log(n) loglog^{1+eps}(n) )$ is, for all $\eps > 0$.

**Questions:**

N/A

**Ethical Concerns:**

["NO or VERY MINOR ethics concerns only"]

**Final Justification:**

I am still in support of accepting this paper after looking at the other reviews (there was not much discussion, since all reviewers are overall supportive).

**Limitations:**

yes

**Quality:**

4

**Strengths And Weaknesses:**

This is a nice, well-written paper that studies a fundamental problem in probability. The paper completely answers the question it sets out to solve: get bounds on the growth rate of the expected value of the \ell-th order statistic. The only weakness is that the scope might be a bit narrow. I.e., this problem is of interest to a small sub-community of NeurIPS, e.g., statisticians, or researchers in auction theory (the expected k-th order statistic is the revenue of a k-position auction). Still, I think that asking that a NeurIPS paper is of interest to everyone who attends NeurIPS is impossible, so I am in support of accepting this work.


======

Here are some typos (not sure where else I should point them out):
Typo in line 54: “ℓ-the” should be “ℓ-th”
Typo in line 55: “Our necessary condition for to have”
Typo in line 156: “escalatorprobability”

---

> ### Author Rebuttal · Authors · 2025-07-30
>
> We appreciate Reviewer WsQo’s positive assessment. We will emphasize the broader applicability of our results and fix the typos noted (lines 54, 55, and 156) in the revised version.

---

### Official Review · Reviewer_L83L · 2025-06-25

**Clarity:** 3
**Significance:** 3
**Originality:** 3
**Rating:** 5
**Confidence:** 3

**Summary:**

The authors consider n iid random variables drawn from a distribution F and are concerned with the problem of the asymptotic (largest possible) grow of extreme order statistics (in the following, I denote with $M_1$ with the max, and $M_n$ with the minimum).
For the maximum it is well-known that $M_1 \in o(n)$ and for any sub-linear function $g(n)$ there exists a distribution $F$ such that $M_1(n) \ge g(n)$.
The authors thus focus on the case $l > 1$. Specifically, they show that for all $l$, the largest possible grow is $O(n/(log(n) log(logn))) and there exists a distribution such that the growth is $n/(log(n) log(logn)))^(1.01)$.
The authors also run experiments which shows how the although the multiplicative gap between the maximum and the second maximum grows quickly with n, the ratio remains approximately constant in 99% of trials.

**Questions:**

Can you comment more on the experiments of Figure 3 and Corollary 1? It seems that the growth of M_2 is approximately $(n/log(n))^{1/2}$ (which contradicts the theory). Is this because the simulated values of n are too small?

**Ethical Concerns:**

["NO or VERY MINOR ethics concerns only"]

**Final Justification:**

The authors addressed my concerns on some (minor) aspects of the technical proofs.

**Limitations:**

Theoretical work. I do not foresee a direct path to potential negative societal aspects.

**Quality:**

3

**Strengths And Weaknesses:**

# Strengths
- This paper fills a well-motivated theoretical problem (see the applications subsection in Section 1)
- The resulting characterization of the extreme order statistics is tight and interesting
- The main text is overall well-written and clear

# Weaknesses
- Not much. Apart from the question below, and some (probably) **errors** that I have found in a proof (which I believe can be fixed without impacting the results). Below, I provide some detailed comments on the proofs together with the details on those errors. **Please, fix them (or correct me) during the rebuttal. For the moment, my score is Borderline Accept, but it is a placeholder. (I will increase it / decrease it based on the output of the rebuttal).**

# Clarity of the appendix
Overall, given the few pages in the appendix, I suggest the authors to provide more explicit steps while deriving their results. While proof-reading the results, I needed to spend some time verifying some (actually many) algebraic steps. Even a small tag near an inequality to justify a step would help with the workload on the review process a lot. I would also encourage the authors to provide more intuition on how the results were derived (i.e., more details on the main intuition behind the proofs).

# Comments on the proofs
### On Lemma 3
- Not really clear to me the comment on line 680

### On Lemma 4
- In the proof of the upper bound, it seems that it is missing an $e^{n/k}$. Indeed, you are only reporting the upper bound on the Poisson. So maybe the result should involve an $e^2$?
- For the LB, it is not clear how you use "for $i$ small enough relative to $n$" (line 688). It seems to me that it might be in the last step to assume that everything is positive, so that you can further proceed by lower bounding the binomial. Also, that last step is not clear to me. Please justify.
- Related to the above. Given the statement of the Lemma, it is not clear that $n$ should be large enough wrt $l$.
- It might be nice to comment a bit in the paper on why a non-trivial UB is needed in Lemma 4, while a trivial UB can be used in Lemma 3

### On Theorem 5
- In the proof of Theorem 5, it seems that the authors are applying Proposition 2 for $F^{-}$. The result, however, is stated and proved only for $F^+$.  Although the upper bound holds, it is not clear how the authors are proceeding for the LB. (p.s., I can see that $F^+$ and $F^-$ are naturally close to each other in shape, and so that similar arguments could hold, in this sense I believe that the result is probably correct, but can you provide a formal argument for this?)
- In Eq. 7, the upper bound does not seem to be needed in the rest of the proof. Right?

### On Lemma 6
- Is there an extra n/x in the equation below line 740?
- The equality after the "Since..." does not hold (line 742). There is an $i$ missing at the numerator so that we have n chooses i * i/n = (n - 1) chooses (i-1).
- I feel that this should not be a problem for the statement, but can you please provide a corrected proof of the statement?

# Other Comments
- Typos: below line 156 "escalatorprobability"
- Line 210, reference missing
- Lines 708, 709 "escalatordistribution"
- Suggestion: In the introduction, stress slightly more that $l$ fixed implies that we have no guarantees, e.g., for the min, as $l$ needs to be constant and hence cannot grow with $n$
- Suggestion: A table somewhere would probably help the reader to maintain a clear big picture, i.e., the existing results + the new results with reference to the various theorems/proposition and their implications
- Sentence at lines 220-221 not clear to me
- Appendix A.9 is never mentioned in the main text
- Proposition 5 states that for any sublinear $g(n)$, there exists a distribution $F$ such that $M^F_l(n) \ge g(n)$ for infinitely many $n$'s. Could you comment more on why this implies the comment on lines 206-208? i.e., the fact that for a fixed $F$ one cannot find a specific sublinear function such that $M^F_l(n) \in o(n)$ (e.g., maybe specifying with words that $g$ cannot depend on $F$ would help the reader)
- Also, for the sake of clarity, it might be useful for the reader to write somewhere the implications of liminf and limsup and how they relate to the asymptotic notation

---

> ### Author Rebuttal · Authors · 2025-07-30
>
> We thank Reviewer L83L for their careful and constructive review. We greatly appreciate the time and effort spent in closely reading our technical proofs and providing concrete suggestions for clarity and completeness. Below, we respond to each technical point in turn.
>
> **1. Comments on Lemma 3 (line 680).**
>
> The comment refers to the fact that the inequality derived so far uses $k\ge 2$, but it is also true for $k=1$.
>
> **2. Comments on Lemma 4 (upper bound).**
>
> Thanks for the correct observation. Indeed, there is an extra $e^{n/k}$ term missing, which should cancel out with the $e^{-n/k}$ from the Poisson distribution, leading to the same upper bound (no extra $e$ term). We apologize for the error, and sincerely thank you for pointing this out.
>
> **3. Lemma 4 (lower bound, line 688).**
>
> Apologies for the confusion. First, note that we substitute $ i=\ell $ in the last step of the proof. Secondly, it is true that we use that $\ell$ is constant (independent of $n,k$), and $n$ is large enough with respect to $\ell$, e.g. $n = \Omega(\ell^2)$. We will add this to the statement of the Lemma as per your suggestion. Indeed, if $n \ge c\ell^2$, then $(1-i^2/n)(1-1/n) \ge (1-1/c)(1-1/(c\ell^2)) \ge (1-1/c)^2$. Note that the term $e^{i/k}$ is at least 1.  Substituting $c=1/(1-\sqrt{1-1/e})$ yields the desired lower bound.
>
> **4. Comment on why a non‑trivial UB is needed in Lemma 4 vs Lemma 3.**
>
> Thanks for the suggestion. The bounds on Lemmas 3 and 4 are related to Theorem 1. Given Proposition 2, for $k<n$ (as in Lemma 3), the Theorem needs only constant bounds on $P(Y_k \ge \ell)$. For $k \ge n$, the Theorem needs tighter bounds, depending on $(n/k)^\ell$, to control the infinite sum.
>
> **5. Theorem 1 (Proposition 2 for $F^-$).**
>
> Thanks again for the suggestion. The result for $F^{-}$ is indeed analogous, and we will add this to the revised version. The main observation is that the definition of $F^{-}$ is the same as in $F^{+}$ but with $\delta_k$’s shifted in their index by one. As a consequence, by shifting indices, the analogous result of Proposition 2 applied to $F^{-}$ yields $M_\ell^{F^{-}} = \sum_{k=1}^{\infty} P(Y_{k+1} \ge \ell) \delta_k$. Applying Lemmas 3 and 4 yields Equation (6) (line 696).
>
> **6. Theorem 1 (Unused upper bound in Eq. (7)).**
>
> Correct. We will avoid this additional fact in the revised version.
>
> **7. Lemma 6 (equation below line 740).**
>
> The equation is correct. After factorizing $(1-1/x)^{n-1}1/x$, the first sum in Equation (9) (with $ i = 0,\ldots,\ell-1 $) is multiplied by $n/x$. In the next equation, we separate the 0-th term, which is $n/x \times 1$, and then factorize the remaining sum (with $ i = 1,\ldots,\ell-1 $) with the second sum in Equation (9), yielding a factor $ n/x - i $.
>
> **8. Lemma 6 (binomial coefficient equality in line 742).**
>
> Apologies for the typo. In fact, the LHS of the equality below that has also the same typo, since the $i$ in the numerator inside the sum should be a 1, following the correct equality. Having this corrected, the RHS follows from an index shift (and is indeed true). The remainder of the proof holds as is. Thanks for the observation.
>
> **9. Typos and missing references in lines 156, 210, 708, 709.**
>
> Thank you for pointing them out. All typos have been corrected in the revised version.
>
> **10. Suggestions for the introduction.**
>
> Very much appreciated suggestions. We will consider them for the revised version.
>
> **11. Sentence at lines 220-221.**
>
> Thank you for the feedback, and apologies for the confusion. We will clarify in the revised version as follows. We can always find probability distributions to make $M_\ell^F(n)$ match any sublinear function infinitely often (for infinitely many $n$). However, if one seeks to find a distribution for which $M_\ell^F(n)$ is large for all $n$, then the threshold lies at [...].
>
> **12. Appendix A.9 is never mentioned in the main text.**
>
> Thanks for the observation. We will correct this in the revised version.
>
> **13. Proposition 5 (lines 206-208).**
>
> Thank you for the suggestion. The (now corrected) comment should specify that one cannot find a specific sublinear function $g$ such that: $M_\ell^F(n) \in o(g(n))$ **for all distributions $F$ with finite expectation**. This follows from Proposition 5, since whenever we fix $g$,  then we can find a distribution $F$ with finite expectation for which the statement $M_\ell^F(n) \in o(g(n))$ is false.
>
> **14. Relation between $\liminf$, $\limsup$ and asymptotic notation.**
>
> Thanks for the observation. We will correct this in the revised version.
>
> **15. Comment on Figure 4 and Corollary 1.**
>
> Corollary 1 states that under finite variance, $M_2$ cannot achieve $\Omega((n/log(n))^{1/2})$. However, this result does not rule out $\Omega(g(n))$ for $g(n)=(n/log(n)^{1+\epsilon})^{1/2}$ with arbitrarily small \epsilon>0. Indeed, the infinite sum $\sum_n g(n)^2/n^2$ can be shown to converge via Cauchy condensation test. If this is the case, the gap can be imperceptible in Figure 4.

---

> > ### Comment · Reviewer_L83L · 2025-08-04
> > **Ack.**
> >
> > I thank the authors for the clarifications. They solved my concerns.
> >
> > I have modified my score to 5 (Accept).

---

### Official Review · Reviewer_91W4 · 2025-07-01

**Clarity:** 4
**Significance:** 4
**Originality:** 4
**Rating:** 6
**Confidence:** 4

**Summary:**

The paper answers a basic question in the asymptotic behaviour of the expectation of the second, and more generally $\ell$’th maximum of i.i.d. random variables.

While the case of the max ($\ell=1$), it is known that the growth rate can be up to $o(n)$ - in the sense that there is a linear upper bound known, but there exists a finite expectation  distribution $F$ such that $M_{1}(n)=\Omega(g(n))$ - the growth rate for $\ell>1$ is tightly characterised as both $O(\frac{n}{\log(n)\log\log(n)})$ and $\Omega(\frac{n}{\log(n)(\log\log(n))^{1.01}})$.

The authors also introduce a reduction to what is referred to as an escalator distribution; a convenient measure discretisation of a a cumulative distribution $F$, the order statistics of which both upper and lower bound the corresponding order statistics of F.

Theory is supplemented with extensive experiments  and characterising behaviour of empirical vs. worst-case separations.

**Questions:**

Questions:
* The escalator distribution reduction is elegant. Is there a possibility of extending this technique to dependent sample scenarios, or is independence fundamentally required for the escalator arguments to hold?
* Given the broad applicability, could you clarify practical guidelines for using your results in robust risk or OOD detection pipelines? Specifically, how might practitioners identify whether their system is in a “regime” where your bounds meaningfully constrain the top $\ell$ scaling - is it as simple as characterising the maximal degree of finite moments?

**Ethical Concerns:**

["NO or VERY MINOR ethics concerns only"]

**Final Justification:**

While it may have been the case, given the other reviewers' identification of typos and points of clarification in the proof, that my assessment was on the generous side initially, I find that against criteria of technical novelty, clarity of exposition (modulo typos), strength of results and breadth of possible applicability, this work still has quite an advantage. The authors' responses to my questions helped elaborate on the breadth of applicability as well as possible extensions of the methods developed. In addition, the extra clarification due to the diligence of reviewers 6S2N and L83L has been instrumental in clarifying its correctness. I recommend acceptance.

**Limitations:**

Yes.

**Paper Formatting Concerns:**

No concerns.

**Quality:**

4

**Strengths And Weaknesses:**

Strengths:
* The paper offers a surprising perspective on the behaviour of the $\ell$’th order statistic for $\ell > 1 $ with extremely general and tight results (up to a minuscule $\log\log(n)^{0.01}$ factor), applying to all distributions with finite expectation.
* The techniques, notably the escalator distribution reduction, are elegant and provide a clean path to bounding asymptotics of order statistics rigorously.
* The applicability of the results is broad, including value-at-risk estimation, option pricing, auction theory, load balancing in compute, and - as noted in Section 1.1 - several high-impact areas in machine learning theory (e.g., adversarial risk, best-arm bandits, OOD detection).
* The paper zeroes in on concrete cases (finite variance, higher moments), illustrating how growth rates degrade systematically under additional distributional constraints.
* Results fill a natural and under-explored gap in the literature between known results for the maximum and the poorly understood behaviour of lower extreme order statistics.

Weaknesses:
- Presentation is somewhat notation-heavy, and highly technical, which may limit accessibility for non-theoretical or applied ML audiences. Considering the theoretical interest of results, this is a rather minor weakness.
* The paper could do more to explicitly connect theoretical findings to practical thresholds, and provide concrete examples like those elaborated in Section 1.1.

---

> ### Author Rebuttal · Authors · 2025-07-30
>
> We thank Reviewer 91W4 for the thoughtful and supportive review. Below, we respond to the questions raised and provide clarifications where requested.
>
> **1. Escalator distribution and independence assumption.**
>
> This is an excellent question. Our current argument relies critically on independence (e.g. Proposition 2) and does not generalize directly to dependent sample scenarios. Extending the escalator reduction to dependent settings would require fundamentally new ideas and is an interesting avenue for future research.
>
> **2. Practical guidelines for robust risk and OOD detection.**
>
> At a high level, our results highlight that practitioners should be especially cautious about the behavior of the maximum vs other aggregation methods. In OOD detection, for example, it is common practice to detect outliers through thresholds on the maximum of confidence scores. In the heavy‑tailed regimes we study, the maximum sample can be orders of magnitude larger than the second maximum, a phenomenon our bounds characterize and which, importantly, can occur even when all samples are drawn from the same distribution (with finite mean). Our results also relate to the use of top-$\ell$ aggregate loss in robust learning (see e.g. Fan et al., NeurIPS 2017). Indeed, under heavy-tailed distributions, the largest individual loss might dominate the average top-$\ell$ aggregate loss when the number of training examples is large, potentially undermining its intended robustness.
>
> References:
>
> Y. Fan, S. Lyu, Y. Ying, and B. Hu. Learning with average top-k loss. In Advances in Neural Information Processing Systems 30, pages 497–505, 2017.

---

> > ### Comment · Reviewer_91W4 · 2025-08-05
> >
> > I thank the authors for their detailed answers to my questions (and the other reviewers for spotting typos that I missed, and thereby allowing their resolution during the rebuttal phase). I maintain my score.

---

### Official Review · Reviewer_W6kr · 2025-07-03

**Clarity:** 3
**Significance:** 3
**Originality:** 3
**Rating:** 5
**Confidence:** 4

**Summary:**

This paper studies the worst-case asymptotics of extreme order statistics, e.g. $M_{\ell}(n)$  the $\ell$-th greatest value among $n$ i.i.d. positive random variables, for a fixed $\ell$ as $n$ goes to infinity. For $M_1(n)$, is has been previously shown that $M_1(n)=o(n)$, and $\liminf M_1(n)/g(n)>0$ for any $g(n)$ sublinear in $n$ tightly characterizing the worst-case asymptotics. Contrasting with this previous result, this papers shows that while $\limsup M_{\ell}(n)/g(n) >0$ remains true for all sublinear functions $g(n)$, it does not necessarily hold anymore for $\liminf$. In particular, it is shown that $M_{\ell}$ consistently dominates $g(n)$ if and only if $\sum_n g(n)/n^2<\infty$, providing a complete picture on the case $\ell>1$. Additional results are provided when additional moments are finite.

**Questions:**

- By the fact that the threshold $g(n)$ does not depend on $\ell$, does this suggest that all of the $M_{\ell}$ for $\ell>1$ have all the exact same worst-case asymptotic behavior? Why is that the case?
- If we have that $\sum_n g(n)/n^{\ell} <\infty$, can we say anything more about $M_{\ell}$?
- What can we say about the asymptotics of order statistics gaps $M_{\ell}-M_{\ell+k}$?
- line 679, why can we upper bound $1/(k-1)^i$ by $1/(n-1)^i$ given that $k\leq n$? I must have missed something obvious.

There are some small typos to correct:
- line 55 Conditions for to have
- line 221 There exists infinite ... sublinear functions. (confusing sentence)

**Ethical Concerns:**

["NO or VERY MINOR ethics concerns only"]

**Final Justification:**

This is a technically solid paper, with novel contributions, and while I was not convinced by the broader applications mentioned, this problem is still well motivated by itself.

**Limitations:**

Yes.

**Paper Formatting Concerns:**

None.

**Quality:**

3

**Strengths And Weaknesses:**

## Strengths

- It provides a comprehensive theoretical analysis of the extreme order statistics, and highlights a very non-intuitive subtle difference in asymptotic behavior between $M_1$ and $M_{\ell}$ for $\ell>1$.
- The paper is clearly written and the related works are precisely presented.

## Weaknesses

- Given that this paper is based on a quite technical aspect of limit behavior of order statistics, it would be better to be as unambiguous as possible on the meaning of the $\Omega(g(n))$ and other Landau notations. In some fields, these notations sometimes implicitly assume that the ratio $f(n)/g(n)$ converges, which is clearly not the case here as $\limsup$ and $\liminf$ are distinct.
- The paper is lacking a bit in terms of intuition in why there is such a difference between $M_1$ and $M_2$, and why there is a difference in terms of $\limsup$ and $\liminf$. Why does this phenomena not appear at $\ell=4$ for instance?
- The paper mentions potential applications, and while I really appreciate the result, I still struggle to see how this would concretely impact existing bounds or results in other topics. It would be great to highlight on a small example possible consequences, for instance to detail how the the main result concretely affects BAI (being one of the example suggested).
- This paper would benefit from additional visual aids, such as plotting the escalator transformations and the worst case distributions. It might be a good use of the additional space to add these plots.

---

> ### Author Rebuttal · Authors · 2025-07-30
>
> We thank Reviewer W6kr for the valuable feedback and suggestions. Below, we respond to each specific question.
>
> **1. Threshold $g(n)$ and dependence on $\ell$.**
>
> We think of $g(n)$ as an exogenous benchmark, and then analyze if $M_\ell^F(n)$ can achieve such asymptotic growth rate under some distribution $F$, and whether this answer depends on $\ell$. Indeed, the answer is the same for any constant (w.r.t. $n$) $\ell \ge 2$ (Theorem 2). The main technical reasons for this result can be summarized as follows. If we fix $g(n)$ satisfying $\sum_n g(n)/n^2<\infty$, then for any constant (w.r.t $n$) $\ell \ge 1$, we can construct a tailored distribution $F$ to achieve $M_\ell(n) = \Omega(g(n))$, and of course $M_1(n) \ge M_2(n) \ge \ldots \ge M_\ell(n)$. For the other direction, we show that if $M_\ell(n) = \Omega(g(n))$, then $\sum_n g(n)/n^2 \le A (1+\sum_{k>n}\frac{n^{\ell-1}}{k^\ell})$, where $A$ is close to the distribution mean (finite). If $\ell=1$, this upper bound is infinite (uninformative), but for any $\ell \ge 2$, the infinite sum is bounded by a constant.
>
> **2. Condition $\sum_n g(n)/n^\ell < \infty$.**
>
> Aside from the special case $\ell=2$, this condition is not very informative: the sum diverges for $\ell=1$, and converges for $\ell\ge3$. The case $\ell=2$ is the only one where more nuanced behavior arises.
>
> **3. Asymptotics of order statistic gaps $M_\ell - M_{\ell+k}$.**
>
> For $\ell = 1$, our results imply that the gap can be very large, growing almost linearly in $n$ (even if $k=1$). For $\ell\ge1$, we performed additional simulations and find that the gap is significantly smaller than the $\ell=1$ case.
>
> **4. Line 679: upper bound justification.**
>
> The bound applies to the entire sum, which Lemma 6 (Appendix A.8) shows to be non‑decreasing in $k$.
>
> **5. Typos.**
>
> Thank you for pointing them out. All typos have been corrected.

---

> > ### Comment · Reviewer_W6kr · 2025-08-03
> >
> > I thank the authors for their replies and clarifications.

---

### Decision · Program_Chairs · 2025-09-17

**Decision:**

Accept (poster)

**Comment:**

This paper studies the growth rate of the expectation of general $\ell$-th order statistics with respect to the number of i.i.d. samples. This paper derives a tight characterization of the growth rate, which demonstrates the significant difference from the case of the maximum ($\ell=1$). The findings are also validated through numerical simulations.

The reviewers agreed in the opinion that the paper studies a very fundamental problem and the derived result is significant. Though various technical concerns are raised from L83L and 6S2N, they are clearly solved by the rebuttal. I expect that these points and other comments on the presentation are carefully addressed in the final version.

Remaining concern is that the result may be of limited interest for NeurIPS community and the practical usage is somewhat unclear. I also agree with this point. In fact, the distribution giving the bound in this paper has extremely heavy tails, and I'm not sure how it is relevant for practical scenarios. For example, I don't think that such distribution appears in the reasonably-designed second-price auction. While the reviewers and I agree that this point is not serious, this is part of the reason why I do not recommend spotlight presentation even though I personally highly enjoyed the paper.